

# Foehn effect during easterly flow over Svalbard

Anna A. Shestakova[1], Dmitry G. Chechin[1], Christof Lüpkes[2], Jörg Hartmann[2], Marion Maturilli[2]

[1] Air-Sea Interaction Laboratory, A.M. Obukhov Institute of Atmospheric Physics of the Russian Academy of Sciences, Moscow, 119017, Russian Federation

[2]Alfred Wegener Institute, Helmholtz Centre for Polar and Marine Research, Bremerhaven (Potsdam), Germany

*Correspondence to*: Anna A. Shestakova (shestakova@ifaran.ru)

**Abstract.**

This article presents a comprehensive analysis of the foehn episode which occurred over Svalbard on 30-31 May 2017. This
episode is well documented by multiplatform measurements carried out during the ACLOUD/PASCAL campaigns. Both orographic wind modification and foehn warming are considered here. The latter is found to be primarily produced by the isentropic drawdown, which is evident from observations and mesoscale numerical modelling. The structure of the observed foehn warming was in many aspects very similar to that for foehns over the Antarctic Peninsula. In particular, it is found that the warming was proportional to the height of the mountain ridges and propagated far downstream. Also, a strong spatial
heterogeneity of the foehn warming was observed with a clear cold footprint associated with gap flows along the mountain valleys and fjords. On the downstream side, a shallow stably-stratified boundary layer below a well-mixed layer formed over the snow-covered land and cold open water. The foehn warming downwind Svalbard strengthened the north-south horizontal temperature gradient across the ice edge near the northern tip of Svalbard. This suggests that the associated baroclinicity might have strengthened the observed northern tip jet. Positive daytime radiative budget on the surface, increased by the foehn
clearance, along with the downward sensible heat flux provoked an accelerated snowmelt in the mountain valleys in Ny-Alesund and Adventdalen, which suggests a potentially large effect of the frequently observed Svalbard foehns on the snow-cover and the glacier heat and mass balance.

## 1 Introduction

The observed climate change in the Arctic is strong, while the mechanisms are not yet fully understood (Serreze and Barry, 2011; Dethloff et al., 2019). Svalbard is also experiencing climate change, which manifests itself in warmer winters (Dahlke and Maturilli, 2017), shorter snow cover duration (Descamps et al., 2017), larger area of the Whaler's Bay polynya north of Svalbard and increased air temperatures over Fram Strait during cold-air outbreaks (Tetzlaff et al., 2014), negative trend in glaciers mass-balance (Hagen and Liestol, 1990; Hagen et al., 2003; Nuth et al. 2010) and reduced sea ice cover in the fjords
(Zhuravskiy et al., 2012; Muckenhuber et al., 2016; Dahlke et al., 2020). Altogether this results in the fact that the highest rate of temperature rise in Europe is observed in Svalbard (Nordli et al., 2014).



The climatic signal over Svalbard is strongly modulated by the orography (Beine et al., 2001; Argentini et al., 2003; Kilpeläinen and Sjöblom, 2010; Kilpeläinen et al., 2011). To better extract the climatic signal from the observational time series, one needs to study the orographic effects and their dependence on the parameters of the large-scale flow. The focus of this study is on

the orographic winds observed during foehn, namely, downslope windstorms, gap flows and tip jets, and their impact on the lower atmosphere and the surface heat budget.

There are several reasons why it is important to investigate these winds in more detail. First, Fram Strait is one of the "hot" spots of the Arctic climate system: Atlantic water experiences transformation as it inflows into the Arctic, while sea ice is being exported through Fram Strait, where it melts, or forms (during cold-air outbreaks). These processes can be affected by

the orographic winds due to the influence of the latter on the momentum and heat transport between atmosphere and ocean as well as on the heat budget of the sea-ice surface. This is especially the case for foehn since the foehn warming can propagate far from the mountains (Elvidge and Renfrew, 2016). The impact of orography is also strong in fjords where it influences the energy exchange between the often ice-free sea surface and the atmosphere. In fjords, wintertime cooling is driving dense water mass formation, which might be important for the regional ocean circulation (Skogseth et al., 2007). According to model

results (Kilpelainen et al., 2011), values of turbulent heat fluxes can reach up to 500 W m$^{-2}$ for sensible heat, and 300 W m$^{-2}$ for latent heat over the wintertime ice-free sea surface in Isfjorden (Svalbard), even during moderate easterly foehn winds. This is due to large temperature and humidity differences between the open sea surface and the advected cold airmass.

Another crucial component of the climate system, which is affected by the orographic winds, is the energy exchange between the atmosphere and glacier surfaces. Svalbard glaciers serve as indicators of the amplified Arctic climate change. The glacier

mass balance on Svalbard is the most negative in the Arctic (Nuth et al., 2010). Foehns are known to have an impact on the heat and mass budgets of glaciers. A high frequency and intensity of foehns can lead to an increased melting of glaciers. This is well documented for Antarctica, where foehn has a strong effect on the mass balance of glaciers and sea ice in the Antarctic Peninsula region (Elvidge et al., 2015).

Finally, orographically induced winds and turbulence are often strong and represent danger for aviation and other human

activities. Heterogeneity of the orographic winds, i.e. the occurrence of strong jets and wakes, can be so strong in the Svalbard fjord areas that it poses danger to small vessels (Barstad and Adakudlu, 2011). Also, it is documented that downslope windstorms to the west of Svalbard cause ship icing (Samuelsen and Graversen, 2019).

Such strong windstorms frequently occur on the western slopes of Svalbard with a prevailing large-scale easterly flow and are mentioned in many articles (e.g., Kilpelainen et al., 2011; Mäkiranta et al., 2011; Beine et al., 2001; Maciejowski and

Michniewski, 2007; Migala et al., 2008; Láska et al., 2017). However, downslope windstorms over Svalbard and especially the foehn effect have not yet been well documented. Mostly, only the near-surface observations and the associated large-scale circulation are discussed in relation to the Svalbard foehns, while the three-dimensional analysis of this phenomenon is missing.

In addition to downslope windstorms, other local winds are observed over Svalbard - glacier winds (Esau and Repina, 2012),

gap flows and tip jets. Gap winds and glacier winds are often observed together and are directed along the fjords. Gap winds





usually spread far from the obstacle, if the gap is not too narrow and the Froude number is not too small (Gaberšek and Durran, 2004). Tip jets at the southern and northern edges of the archipelago can be quite strong (Reeve and Kolstad, 2011; Sandvik and Furevik, 2002) and lead to significant changes in heat fluxes over the sea surface, similar to the well-known southern Greenland tip jet (Doyle and Shapiro, 1999).

The main difference between the listed winds lies in their nature, i.e., in the physical mechanism of their formation. Under the action of a pressure gradient, air flows into the gaps (during gap winds) or around an obstacle (during tip jets), stream lines converge and velocity increases (Markowski and Richardson, 2011). When the air at least partially overflows the obstacle, a downslope windstorm may occur. If this is the case, convergence of stream lines on the leeward side occurs due to i) the transition of the flow to the supercritical state and/or due to ii) breaking of high-amplitude internal gravity waves over the

mountains. Downstream, a hydraulic jump (a jump-like change of the flow thickness) usually occurs and the flow becomes subcritical, where a calm zone (wake) is formed (Skeie and Gronas, 2000). Glacier, or katabatic, winds are produced by the cooling of air over a sloping glacier surface which results in the horizontal pressure gradient along the slope. Despite the existing knowledge on the orographic flows, their local features need to be studied for each particular region because the complex local orography and surface conditions may lead to a co-existence and interaction of several flow types and regimes.

Foehn winds are particularly important because they are associated with two effects: i) downslope wind storm and ii) foehn warming. Numerous studies of foehns in different regions have focused primarily on the dynamics of the downslope winds (e.g., Brinkmann, 1974; Hoinka, 1985; Skeie and Grønås, 2000; and many others). However, Elvidge and Renfrew (2016) and Elvidge et al. (2016) draw attention to the detailed structure of foehn warming downwind the Antarctic peninsula and its effect on the ice shelf melt. Previous investigations of foehns also addressed this issue (e.g., Seibert, 1990; Olafsson, 2005; Steinhoff,

2013) but were not as detailed and comprehensive. Elvidge and Renfrew (2016) list the following mechanisms which produce the foehn warming: 1) isentropic drawdown; 2) turbulent sensible heating (mechanical downward mixing of warmer air in the stratified flow); 3) radiative heating (due to the cloud-free conditions on the lee side of the mountains); and 4) latent heat release and precipitation mechanism. They showed that the prevailing mechanism depends on the Froude number of the flow (i.e. wind speed and stratification) as well as the moisture content of the incoming flow and that several mechanisms can act

simultaneously. The goal of our study is also to document the structure of foehn warming and its effect on the surface heat balance, however, in a region with a more complex orography as compared to the Antarctic Peninsula. The latter is called by Elvidge et al. (2016) «an excellent natural laboratory» to study foehns due to the almost two-dimensional structure of the mountain ridge there.

Thus, the goal of this paper is to document in detail the orographic modification of the flow and of the atmospheric boundary

layer during easterly flow over Svalbard. To that aim, we used a unique set of observations on 30-31 May 2017 collected during the aircraft campaign **A**rctic **CL**oud **O**bservations **U**sing airborne measurements during polar **D**ay (ACLOUD) and the shipborne campaign **P**hysical feedback of **A**rctic PBL, **S**ea ice, **C**loud **A**nd Aeroso**L** (PASCAL) (Wendisch et al. 2019). Also, we used observations from Ny-Alesund, Svalbard, where the frequency of radiosoundings operated by the AWIPEV research base was increased to four launches per day in connection to the ACLOUD/PASCAL campaigns. Using this set of





observations, we aimed to gain better understanding of the strength and structure of the orographic modification of the flow. We also used high-resolution simulations of the Weather Research and Forecasting (WRF) model to obtain a better three-dimensional view of the orographic impact on the flow.

The structure of the paper is as follows: the used observations are described in Section 2.1. The setup of numerical simulations with WRF is presented in Section 2.2, synoptic background of the considered episode is described in Section 3.1 and in Sections 3.2-3.4 the structure of the orographic winds and foehn warming is described and analyzed. In Section 3.5, the effect of foehn on the surface heat budget over the snow-covered surface is considered. The main results are summarized in Section 4.

## 2. Data and methods

### 2.1 Observations

We investigated a period of increased wind velocities and air temperature over the western part of Svalbard, which occurred on May 30–31, 2017 during large-scale easterly flow. Complex observations in the boundary layer were performed during this episode, including in-situ surface based observations, aircraft and ship observations, vertical profiles obtained from drop sondes and radiosondes launched from ship and over land (in Ny-Alesund). Automatic weather station (AWS) data was available in addition to standard observations at Ny-Alesund, Longyearbyen and Barentsburg stations. Figure 1 shows only those stations whose observations were used in this article. Figure 1 also shows the track of the research vessel (RV) Polarstern (Knust, 2017; Wendisch et al., 2019) and the locations of the ship radiosonde and aircraft drop sonde launches as well as the aircraft tracks (aircraft Polar 5 and 6). The information about the used observations is summarized in Table 1.

The two research aircraft of the German Alfred Wegener Institute (AWI), Polar 5 and Polar 6, were involved in the ACLOUD campaign. Identical sets of meteorological sensors were installed on both aircraft, including 5-hole probes and open-wire platinum temperature sensors for turbulence observations (Ehrlich et al., 2019b). Both aircraft conducted their flights on 31 May during the foehn episode. They performed series of ascents and descents through the boundary layer on the downwind and thus western side of northern Svalbard, as well as vertical stacks of horizontal legs over the sea ice in the Fram Strait. Only the vertical profiles of temperature, humidity, wind speed and direction obtained from the aircraft ascents and descents were used in this study. A more detailed description of metadata and summary on the aircraft and shipborne observations can be found in Wendisch et al. (2019) as well as Ehrlich et al. (2019b), and a description of the synoptic situation during ACLOUD is given in Knudsen et al (2018).

To evaluate the foehn effect on the surface heat budget we used measurements of wind speed and direction, air temperature and relative humidity, which are available from two masts with instrumentation at 2 m and 10 m above the ground level (AGL) (Table 1). One of the masts is along Kongsfjord located on the Ny-Alesund measurement field and another one in Adventdalen (station of the University Centre in Svalbard, http://158.39.149.183/Adventdalen/index.html). The observations of both masts are representative for conditions in fjord valleys. These valleys have approximately the same width and are elongated from north-west to south-east.



The YOPP analysis of ECMWF (European Centre for Medium-Range Weather Forecast) (https://apps.ecmwf.int/datasets/data/yopp/levtype=sfc/type=cf/) with spatial resolution of 0.25º was used for the analysis of the synoptic situation during the studied foehn episode.

## 2.2 Numerical modelling

The considered foehn episode was simulated using the mesoscale model WRF-ARW version 3.4.1. The model settings and the domain configuration were the following. We used three nested domains (with two-way nesting): domain 1 with 80x80 nodes and grid spacing of 20 km, domain 2 with 181x201 nodes and grid spacing of 4 km, and domain 3 with 226x457 nodes and grid spacing of 1.3 km (see Fig.2). Such a grid spacing in domain 3 allows foehn to be adequately reproduced in areas with complex topography, as shown by Elvidge et al. (2015). The number of vertical levels was 40 with a vertical grid spacing of about 150 m in the lowest 1 km. Radiative transfer was parameterized using the RRTMG scheme (Iacono et al., 2008). Vertical turbulent transfer was parameterized using the MYNN 2.5 scheme (Nakanishi and Niino, 2009). The simulations used the Noah land surface model and the Kain-Fritsch scheme for convection (only in the outer domain). The Global Forecast System (GFS) final analysis (FNL) with 1˚ resolution was used as initial and boundary conditions. The model was initialized at 30 May 00 UTC and the numerical experiment lasted over 54 hours, till 1 June 06 UTC. The output interval was set to 1 hr.

## 3 Results

### 3.1 Synoptic background

Easterly and north-easterly winds are rather common over Svalbard. Most cases of strong easterly winds over Svalbard are caused by Atlantic cyclones, when they become stationary to the West of Svalbard or when they are moving from West to East across the Barents Sea (Migala et al., 2008; Shestakova et al., 2020). The episode on May 30-31, 2017 occurred in a rather unusual synoptic situation, described in detail in Knudsen et al. (2018). A combination of the cyclone over the Barents Sea (which moved there from the Kara Sea) and an anticyclone over Greenland led to an intensification of the north-easterly flow over the archipelago. The advection of a relatively warm air mass occurred in the northern part of the cyclone (Fig.3). The air temperature at 850 hPa exceeded 0°C upwind of Svalbard, however, it will be shown further that the near-surface temperature there was below zero (i.e. in equilibrium with the cold underlying surface) and the atmospheric boundary layer (ABL) was capped by an inversion.

The considered episode was preceded by relatively cold weather marked by several episodes of an off-ice northerly flow (Knudsen et al., 2018). In this preceding period, the air temperature in Ny-Alesund oscillated around -5 °C (Fig. 4a). On May 29, an abnormally warm period began, when the air temperature was higher than its long-term mean (Knudsen et al., 2018).



With a strong easterly and north-easterly wind (Fig.4b), the air temperature steadily increased and reached the maximum of about 7 °C on May 31. In the beginning of the episode, the vertically integrated water vapor was rather high due to the preceding large-scale moisture advection (Knudsen et al., 2018). Later on May 30, it began to decrease along with the relative humidity, reaching its minimum on May 31 in the morning. This humidity decrease was associated with the change in large-scale flow

direction from easterly to north-easterly (see Fig.B1d) and the advection of a less humid air mass. Also foehn might have contributed to the decreased humidity, as is usually observed during foehns (e.g. Hoinka, 1985; Siebert, 1990; Elvidge et al., 2016).

Wind speed near the surface was rather high as compared to average conditions, but lower than during some documented downslope windstorms (Shestakova et al., 2018; Shestakova et al., 2020). 10-min averaged wind speed attained the values of

15-20 m s$^{-1}$ at 10 m AGL and gusts attained 24 m s$^{-1}$ at the Adventdalen and Hornsund stations. In Ny-Ålesund, the wind speed did not exceed 9 m s$^{-1}$ which is less than 50 percentile of the wind speed during windstorms at this station (Shestakova et al. 2020). Significant fluctuations («gustiness») of all meteorological parameters were evident from measurements with high temporal resolution on the Polarstern (Fig.5) and at the Adventdalen station (not shown) and were especially pronounced in temperature and humidity. Such fluctuations are typical for downslope windstorms (e.g., Klemp and Lilly, 1975; Belusic et

al., 2004; Efimov and Barabanov, 2013; Shestakova et al., 2018).

## 3.2 Spatial distribution of foehn warming

At first sight, the observed warming and air drying in Ny-Alesund and downwind (at the Polarstern location) (Fig.4,5) could

be explained by the large-scale advection of a warmer and drier air mass. However, a closer examination of temperature (relative humidity) observations from different locations pointed out that the large-scale synoptic near-surface warming (drying) was rather weak, but it was amplified by the orography at the locations downwind from the major mountain ridges. Namely, we considered the stations not influenced by the orography, which are Sorkapp (situated on a rather flat island, at 10 m above mean sea level (AMSL)), Kvitoya (situated on the shore of Kvitoya island with a gentle terrain, at 10 m AMSL) and

Verlegenhuken (coastal station surrounded by a flat terrain, at 8 m AMSL) (see Fig. 1). Kvitoya is located upwind the archipelago, while Sorkapp and Verlegenhuken are located at the southern tip and at the northern coast of Svalbard, respectively.

Figure 6 shows the temperature rise, which is the difference between the observed temperature and the temperature at the initial time (00 UTC 30 May), at these stations. Here, we used the temperature rise as a measure of the foehn warming as it

allowed us to estimate in a simple way the warming from the ground-based observations. The drawback of this metric is that it does not accurately distinguish between foehn and non-foehn (air-mass advection, local radiative heating etc.) warming. Note that Elvidge and Renfrew (2016) used a different measure of the foehn warming. Namely, they define the latter as the difference of temperatures at the same height downwind the ridge and in the undisturbed flow upwind the ridge.



A very modest warming reaching only 2°C at its maximum was observed at the southernmost Sorkapp station. Almost no warming was observed also at Kvitoya and Verlegenhuken, as can be clearly seen from Figure 7a. The latter figure shows the maximum temperature rise during the foehn episode.

On the contrary, the stations on the downwind side of Svalbard showed a much stronger warming. Figure 6 shows that in Pyramiden (20 m AMSL), the temperature rise reached 9°C, which is 5-8 times greater than the temperature rise at the Sorkapp station. Obviously, this can only by attributed to the foehn effect. Some stations showed a moderate temperature increase, such as the Akseloya station (20 m AMSL). There, the temperature rise amounted to about 4-5 °C (Fig.6). A weaker temperature rise at Akseloya (Fig.6) could be due to two reasons: first, there was a weaker foehn effect upwind from this station where the flow was passing through the gap along the Bellsund fjord (where Akseloya is located); second, there is a noticeable distance from the nearest upwind mountain ridge and the foehn warming could decrease due to the heat exchange with the colder surface of the fjord. Interestingly, the greatest temperature rise in Pyramiden was accompanied by the lowest wind speed among other stations (Fig.7b). The latter is associated with a wake formation. The distribution of the relative humidity decrease is not shown but this decrease was maximal at the stations with the maximal temperature rise, i.e., at the stations influenced by foehn.

Figure 8 shows that the differences in the magnitude of the foehn warming on the leeward side can be explained by the differences in the nature of the two observed orographic winds: the downslope windstorm and the gap wind. Vertical cross-sections are shown for the gap flow in Bellsund fjord (Fig.8 c,d) and for the downslope windstorm that occurs next to this fjord (Fig.8 e,f) over the southern Svalbard. Concerning the downslope windstorm, a jet stream (area of high wind speed near the foots of lee slopes) descended almost to the surface (Fig.8 e). The height with the maximum wind velocity is 150-200 m and the thickness of the jet was about 500 m, which is consistent with radiosoundings in Ny-Alesund and aircraft in-situ measurements in Longyearbyen regions (not shown). An area of low winds occurred above the jet, most likely associated with gravity wave breaking. This can be seen from the local instability that occurred here (in Fig.8e, the red dots indicate regions where the Richardson number is smaller than 0.25). Jet streams associated with the downslope winds did not propagate far from the lee slopes (Fig.8 a,e). Wind speed near the surface sharply attenuated, and the wake formed in the boundary layer. This wind stagnation in the wake can be clearly seen from Polarstern data and was well reproduced by the WRF model (Fig.5). The abrupt wind speed change was due to the so-called hydraulic jump (Fig.8e). Beneath the hydraulic jump, local instability occurred (Fig. 8e), which indicated the so-called boundary-layer separation (Markowski and Richardson 2011). According to long-term observations (Maturilli et al. 2013), the Ny-Alesund site at the western coast of Svalbard is characterized by the predominance of low wind speed (partly due to the wake formation), although the whole Atlantic sector of the Arctic is characterized by high wind speed (Hughes and Cassano 2015). Combination of high-speed areas near the mountains and wakes downstream is typical for other downslope windstorms, e.g. for the Novaya Zemlya bora (Efimov and Komarovskaya 2018) and Novorossiysk bora (Shestakova et al. 2018), which is not the case for gap flows. The latter propagated far into the sea with wind speed gradually decreasing (Fig.8 a, c). The vertical cross-section of the gap flow (Fig.8c) shows that wind amplification



in them was rather small as compared to the incoming flow and the vertical air displacement was small too. The latter can be seen from the small difference in height of the lower isotherms between windward and leeward sides of the mountains.


Although gap winds on Svalbard had a similar vertical scale and magnitude as downslope winds, difference in their spatial and vertical distribution formed spatially heterogeneous structure of the warming. Downslope winds had a warming effect on the entire lower troposphere (Fig. 8f). This was due to the large-scale lowering of the isentropes (the so-called "isentropic drawdown" (Elvidge and Renfrew 2016)) over the Svalbard mountains, while relatively cold air in the lower layers remained

blocked on the upwind side of the mountains. Such a "warm footprint" of downslope winds extended far enough into the sea, as can be seen in Fig. 8b. Warm foehn air reached Polarstern (Fig.5), which passed 50-100 km from the coast. On the contrary, gap winds and tip jets could attain high speed over the sea, but had insignificant effect on the near-surface temperature (Fig.8 d); therefore, gap winds produced "cold footprints" (comparing with ambient air, warmed by downslope winds) on the temperature map (Fig.8b). A similar cold footprint of gap flows was reported earlier by Elvidge et al. (2015). Nevertheless,

adiabatic air heating over the archipelago was present in the gap flow too, but it was much weaker compared to that in the downslope winds and almost does not affect the surface layer (see point 3 in Fig.9). Therefore, it was called a "dampened foehn effect" by Elvidge and Renfrew (2016).

Foehn warming caused by downslope winds was not homogeneous in different parts of the archipelago due to the different

heights of the mountains. According to the WRF modelling, the maximum temperature rise as well as the maximum temperature difference between the downstream and upstream sides of the archipelago (up to 10 ˚C) occurred over north-western Svalbard (Fig.7). This is obviously due to the larger number and larger height of mountain ridges there compared to southern Svalbard. In the northern part, the mountain ridge heights reach up to 1000-1300 m, while in the southern part they reach only 500-800 m. The presence of a warmer air mass aloft is clearly seen in the model's inflow profiles (Fig. 9) where

an elevated inversion was present with a base at about 700 m height, which is below the mountain top height. As a result, warm air from the inversion layer descended to the surface on the leeward side experiencing adiabatic heating (mechanism №1 from the Introduction) as shown by the modelled profile 2 in Fig. 9. Vertical cross-sections of potential temperature across the northern (not shown) and southern (Fig.8) Svalbard revealed that the air near the surface on the lee side of mountains originated from the height approximately corresponding to the maximum mountain height (if we assume that the isentropes

coincide with the streamlines). The difference in potential temperature between the maximum mountain height and the earth surface in the incoming flow (profile 1 in Fig.9) corresponds to a temperature rise due to the isentropic drawdown (Elvidge and Renfrew 2016). It amounted to about 7-10 K for the northern Svalbard and to about 2-4 K for the southern Svalbard, which is close to the temperature rise shown in Fig.7. Another possible and additional process is the increased turbulent mixing (mechanism number 2 in the Introduction) over the complex orography of northern Svalbard transporting the warmer air

downwards. Radiative air heating (mechanism number 3 in the Introduction) could also have occurred in our case. Starting from the afternoon May 30, there were no clouds on the lee side of the mountains (as observed from the satellite images, see





Fig.1) due to foehn clearance. According to the WRF simulation, precipitation on the windward side of the mountains appeared only in the very beginning of the episode and its amount was negligible; therefore, we conclude that the contribution of latent heat release mechanism (number 4 in the Introduction) was small in the studied case.


### 3.3 Boundary-layer structure

In this section, we investigated the observed temperature structure of the lower troposphere during foehn in more detail with a special focus on the atmospheric boundary layer response to the foehn warming. Namely, we compared the observed and simulated vertical temperature profiles downwind of Svalbard, which were affected by foehn with those in the undisturbed
275 flow upwind of Svalbard and over the sea ice to the north of Svalbard.

First of all, let us consider the east-west potential temperature cross-section (i.e. along the trajectories) during foehn. Figure 9 and 10 show the potential temperature profiles observed (or simulated) upwind, over Svalbard and over water downwind. The simulated upwind profiles were well mixed up to the height of about 600 m with a mixed-layer temperature of about 269.5 K. Over Svalbard, as observed by the radiosoundings in Ny-Alesund, a much warmer mixed layer formed during the considered
280 foehn event. Its height varied from about 700 to 500 m and the mixed-layer temperature was about 276 K. A more detailed analysis of the profiles over Ny-Alesund shows that the lowermost 100 m-thick layer was stably stratified. This is not surprising because the land surface was covered with snow whose temperature was at the melting point, i.e. colder than the overlying air. Surface-layer observations at Ny-Alesund also confirmed stable stratification resulting in a downward (negative) turbulent heat flux, as shown further in Section 3.5.

285 Further downwind, over the open water, the whole air column and especially the lowest 300-400m were stably stratified (Fig.10b). The potential temperature at 500 m height is 285 K, i.e. much warmer than over Ny-Alesund and upwind of Svalbard, while the surface temperature of open water was close to 275-277 K according to the YOPP analysis of ECMWF. In the lowest 500m the low-level temperature inversion was very strong with a vertical gradient of potential temperature of about 1.5 K per 100 m according to the dropsonde data.

290 Figures 9 and 10 also show potential temperature profiles before and during the foehn episode. An enhanced low-level warming during foehn and a transition from a well-mixed to a more shallow and stably stratified boundary layer was especially evident over water and ice edge downwind northern Svalbard.

More detailed observations of the vertical thermal structure downwind of northern Svalbard were obtained during the ascents and descents of P6 along its track towards North (Fig. 11). The profile over the open water T2, which is closest to the downwind
295 side of the mountains, clearly shows the downward propagation of a warm and dry air from aloft to the heights of about 70 m. The observed air temperature at that height reached 10ºC. Below that height a very shallow and strongly stable boundary layer formed over cold water. Further north and away from the mountains both wind speed as well as the height of the stable boundary layer (profiles T3 and T4) increased. Over the rough sea ice, a cold, moist and well-mixed boundary layer was observed with its height increasing to the value of about 300 m in the North (profiles T6-T8). The potential temperature in the
300 mixed layer over sea ice decreased to about -7ºC and was in equilibrium with the sea ice surface temperature. The mixed layer



was capped by a strong inversion with a temperature jump of about 10K. Unlike directly downwind the mountains, where the foehn effect was evident, it is hard to conclude whether this warm and dry air above the boundary layer over ice was solely of advective origin or to some extent also affected by foehn.

The described temperature profiles (Fig.11a) shows a strong north-south horizontal temperature gradient in the lowest layer.
Although the horizontal temperature gradient was mostly due to the ice-water surface temperature difference, it was clearly enhanced by foehn. Also, a sloping of the inversion layer was present as seen in a decrease of the boundary-layer height in the North-South direction. Horizontal temperature gradient and sloping inversion across the ice edge are known to produce low-level baroclinicity resulting in low-level jets (Brümmer, 1996; Chechin et al., 2013; Chechin and Lüpkes, 2017) and breeze-like circulations (Glendening, 1992). A low-level jet that might be related to an ice breeze was indeed present in the Polar 6
vertical profiles with the largest wind speed of up to 16 m s$^{-1}$. The increase of the near-surface wind across the ice edge was also observed at Polarstern. It is hard to identify the main mechanism of the low-level jet because also the tip jet was located over the ice edge. Nevertheless, as shown by Chechin and Lüpkes (2017), the increase of temperature and decrease of the inversion height in the north-south direction increases the easterly component of the low-level geostrophic wind. Thus, it might be possible that the observed easterly low-level jet was produced by a combination of the orographic and baroclinic factors.


### 3.4 Orographic wind dynamics

Elvidge et al. (2016) showed that different flow regimes over the Antarctic peninsula produce contrasting structures of a lee-
side warming. Namely, a nonlinear flow regime, characterized by a presence of nonlinear high-amplitude internal gravity waves and their breaking over the mountains, results in strong downslope windstorms and a strong warming right downwind the ridge which does not propagate further downwind due to a hydraulic jump. In a linear regime no hydraulic jump forms and warming is observed over a larger distance downwind the ridge.

We calculated the Froude number $Fr = U/Nh$ (where $U$ – flow velocity, $N$ – Brunt-Vaisala frequency, $h$ – mountain height)
using mean $U$ and $N$ of the incoming flow (details in Appendix B) as a measure of flow linearity. The flow regime is well-described by the linear theory for the Froude number larger than its critical values (usually taken as unity). A transition to the nonlinear regime occurs when the Froude number becomes smaller than critical (Markowski and Richardson, 2011).

The simulated temporal evolution of the orographic winds during the considered episode is shown in Fig.12 together with the corresponding value of the Froude number (Fig.12, center). The variations of the Froude number were associated with changes
in the incoming flow (Fig. B1, Appendix B). At the beginning of the episode, when the incoming flow velocity was high, the elevated inversion was strong, the stratification of the lower layer was close to neutral (Fig. B1, Appendix B), and $Fr$ was close to unity, the flow "easily" went over the obstacle. Downslope windstorm occurred on almost all western slopes of Svalbard, and in the North it even extended for some distance across the sea. The hydraulic jump, which can be identified by the sharp boundary between high-velocity and low-velocity zones, was not very pronounced at the beginning of the episode. Starting





from 12 UTC on May 30 a wake began to form on the lee side of the mountains as the Froude number decreased. Gap winds were most pronounced at low values of *Fr* (18 UTC May 30 - 6 UTC May 31, June 1). According to the simulation results, the northern tip jet strength depended weakly on temporal variations of *Fr*; the tip jet velocity was decreasing during the considered episode due to the decrease of wind speed in the incoming flow (Fig.B1, Appendix B).

Unlike Elvidge et al. (2016), who considered two contrasting foehn cases, one of which was fully non-linear (*Fr* from 0.15 to 0.4), and the other was close to linear (*Fr ~ 1*), both regimes were observed during our episode. These two regimes are: 1) downslope windstorm with a quasi-linear flow (high Froude number), propagating to some distance from the slope, with weakly pronounced gap flows and wake (06-12 UTC May 30), and 2) more amplified downslope windstorms with nonlinear flow (low Froude number), canyon effects and wakes, formed downstream from the hydraulic jumps (starting from 18 UTC

May 30). In general, a decrease of *Fr* during the considered episode led to an increase in the intensity of downslope windstorms and to amplification of gap winds. Wind speed normalized by the incoming wind speed reached 2 on the lee side when *Fr* <0.8 (Fig.12). The incoming flow direction is also a crucial factor for the orographic flow dynamics. Namely, a downslope windstorm in the north-western part of Svalbard occurred only in the beginning of the episode, when the wind direction was from the East and thus perpendicular to the main mountain ridge direction, while later the direction of the flow changed to

North-Easterly (Fig. B1).

During the considered episode we did not find a clear connection between the nonlinearity of the flow and the spread of foehn warming and its intensity. This can be explained, firstly, by a smaller variation of the Froude number of the flow between the regimes as compared to Elvidge et al. (2016), and secondly, by a more complex orography and also a variable wind direction in the incoming flow.


### 3.5 Surface heat budget in Ny-Alesund and Adventdalen

In previous sections we showed that the effect of foehn over West Spitsbergen was pronounced in air temperature, humidity, wind speed and also resulted in the absence of clouds. Thus, one would expect a strong impact of foehn on the components of the heat budget of the snow-covered land and glacier surface. The surface heat budget is expressed as:

$R + H + LE = B$                                                  (1)

R is the radiative budget:

$R = SW \downarrow - SW \uparrow + LW \downarrow - LW \uparrow$                                (2)

where $SW \downarrow$ and $SW \uparrow$ is downward and upward shortwave radiation, respectively, and $LW \downarrow$ and $LW \uparrow$ is downward and upward longwave radiation, respectively.

H and LE are the turbulent sensible and latent heat fluxes, respectively, and B is the residual heat that is equal to the conductive heat flux to the snowpack (or ground) $Q_D$ , which is further consumed by the snow melt $Q_{melt}$ when melting occurs. The latter is expressed as





$$Q_{melt} = L_i \rho_i \frac{\partial h}{\partial t} \qquad\qquad (3)$$

where $L_i$ – specific heat of melting/freezing, $\rho_i$ – snow/ice density, $h$ – snow/ice thickness.


To obtain the budget in Adventdalen and Ny-Alesund LE and H were calculated from hourly averaged meteorological observations at two levels (Adventdalen) or from just one level and from the surface (Ny-Alesund) using the aerodynamic bulk-formulae

$$H = \rho c_p C_H \Delta U\, \Delta T \qquad\qquad (4)$$

$\quad LE = \rho L C_Q \Delta U\, \Delta q \qquad\qquad (5)$

where $C_p$ is specific heat at constant pressure, $L$ is specific heat of vaporization, $\kappa$ is von Karman constant, $\Delta U$, $\Delta T$ and $\Delta q$ is wind speed, air temperature and specific humidity difference between levels with heights $z_2$ (2 m) and $z_1 = z_{0m}$ for $U$ and $z_1 = z_{0t} = 0.1 z_{0m}$ for T and q. The exchange coefficients are expressed using Monin-Obukhov similarity theory as


$$C_H = C_Q = \frac{k^2}{\left[ ln\left(\frac{z}{z_{0m}}\right) - \Psi_M\left(\frac{z}{L}\right) + \Psi_M\left(\frac{z_{0m}}{L}\right)\right]\left[ ln\left(\frac{z}{z_{0t}}\right) - \Psi_H\left(\frac{z}{L}\right) + \Psi_H\left(\frac{z_{0t}}{L}\right)\right]}, \qquad\qquad (6)$$

where $\Psi_{M,H}$ are the universal stability functions for which the form obtained by Grachev et al (2007) is used for stable stratification and the one obtained by Businger et al (1971) and modified by Grachev et al (2000) - for unstable conditions.

Equations 1-6 were applied to the available observations and the results are shown for both stations in Figure 13. Obviously, the surface heat budget was clearly dominated by the shortwave radiative budget (yellow curves in Fig. 13 a-b). The latter had a pronounced diurnal cycle and reached up to about 150 W m$^{-2}$ in Ny-Alesund and 500 W m$^{-2}$ in Adventdalen during daytime. In Ny-Ålesund, the radiation budget was smaller than in the Longyearbyen area primarily due to a much higher albedo (80-90% in Ny-Ålesund and 40-50% in Adventdalen). The reason for this large difference is that during the foehn event, albedo

significantly decreased in Adventdalen (to 10-15%, Fig.13b) because the snow cover was partially eliminated there, as confirmed by a positive surface temperature on May 31 (Fig.13f) which did not occur in Ny-Alesund. It is known from previous studies that low albedo of 10-15% is typical for the period just after the snow melt in Adventdalen (Sjöblom, 2014). In Ny-Alesund, the snow cover did not disappear as in Adventdalen but the albedo of the snow-covered surface decreased to 60-80% during the foehn episode.


The longwave radiative budget (magenta curves in Fig. 13a-b) was strongly negative during the foehn event and amounted to about -75 W m$^{-2}$ at both sites. Such large values are clearly due to the low amount of water vapor in the air (caused by large-scale advection of dry air mass and by foehn drying) and missing clouds.



As expected, turbulent heat fluxes during the foehn episode increased significantly (Fig.13c-d) which is partly a result of the increased wind speed. Stratification of the surface layer was predominantly stable, with the bulk Richardson number $0 < Ri_B < 0.2$ most of the time. Stable stratification was related to the advection of a warm air mass and the strong foehn effect. This is different from winter conditions when stable stratification is produced by the radiative cooling of the surface. Sensible heat fluxes with absolute values of up to 100 W m$^{-2}$ were directed towards the surface, while the latent heat flux up to 230 W m$^{-2}$

in Adventdalen and 50 W m$^{-2}$ in Ny-Alesund was directed towards the atmosphere (Fig.13c,d). The latent heat flux was 2-3 times larger than the sensible heat flux in Adventdalen. This is due to a decreased relative humidity in the surface and boundary layer related to the foehn effect (Fig.13d) and simultaneously high relative humidity at the surface due to snow melting and evaporation of melt water. In Ny-Alesund the relatively small latent heat flux was due to weak wind speed, though the relative humidity lowering in the surface layer was also well pronounced (Fig.13c). The increase of evaporation and latent heat flux

over a snow-covered surface during warm downslope windstorms is also known from other regions (e.g., Golding, 1978; MacDonald et al., 2012; Hayashi et al., 2005; Garvelmann et al., 2017). The daily average evaporation rate in Adventdalen, calculated from Eq.(4), ranges from 1.5 mm/day on May 29 (before foehn) to 3.4 mm/day on May 31 (during foehn), which is one order of magnitude greater than the evaporation rate in Ny-Alesund (Table 2).

Using the calculated turbulent fluxes and the observed radiative budget, we calculated the residual term $B$ (Fig.14b, black line) (Eq. 1). It represents the heat flux related to the melting of snow/ice during ablation when surface temperature is near 0°C (since May 30 in Ny-Alesund and on May 29-30 in Adventdalen) and heat flux to the ground after snow cover disappearance (only in Adventdalen). Positive surface temperature in Adventdalen on May 31 is explained by a partial (but not complete) degradation of the snow cover, which was noted from satellite images and webcams in Longyearbyen. Thus, the surface

temperature, calculated from the upward long-wave radiation flux, represented the average temperature of two surfaces - snow and soil. Therefore, the heat available for melting can also be calculated for May 31 in Adventdalen. At night, $B$ was negative at both stations, while during the day the potential heat available for melting is large, especially on May 31 (Fig.13a,b). The amount of melted snow can be estimated using Eq. 2 and the average snow density before the thawing season, which, according to measurements on Svalbard, is close to 400 kg m$^{-3}$ (Gerland et al., 1999). In Adventdalen, daily average melting rate has

grown from 7.8 (0.4) mm/day of water equivalent (w.e.) on May 29 to 33.2 (8.9) mm/day w.e. on May 31 in Adventdalen (Ny-Alesund) (Table 2). The lower melting rate in Ny-Ålesund is explained, firstly, by a higher initial snow albedo (in Adventdalen, snow albedo was much lower most likely due to the previous melting periods and a more southerly position), and secondly, a significantly weaker wind than in Adventdalen. One should keep in mind that the estimated melting rate in Adventdalen on May 31 should be interpreted rather as the maximum value because, as noted before, the snow cover in Adventdalen strongly

degraded by that day. It is evident that melting rate on May 30-31 was one order of magnitude greater than evaporation rate (Table 2).





Thus, we found out that most of the radiative budget as well as of the turbulent sensible heat flux was consumed on phase transitions of water (snow melting, snow sublimation and evaporation). The amount of the available heat used for snow melt

was partially reduced by the large values of the turbulent latent heat flux. Thereby, foehn apparently has an effect on the water balance, especially in the Adventdalen region. The abundant and rapid snow melting should lead to a sharp increase in runoff. For example, summer melting due to foehns on Svalbard is often known to cause extreme floods on rivers (Majchrowska et al., 2015). On the other hand, intense evaporation should lead to a decrease in runoff. For example, during foehn in the Alps, about half of the melting snow evaporates (Garvelmann et al., 2017). However, in our case, as in some other areas with foehns

(Golding, 1978; MacDonald et al., 2012; Hayashi et al., 2005), evaporation plays a rather small role in the water balance of the snow cover, as it amounted to only 13% of the melted snow in Adventdalen and 8% in Ny-Alesund during the main melting period (May 30-31).

Finally, we can conclude even though the snow melt would have probably occurred even without foehn, due to the synoptic advection of warm air, but it was clearly accelerated by foehn both in Ny-Alesund and especially in Adventdalen. In particular,

enhanced wind speed and air temperature and cloud clearance during foehn led to an increase in the surface radiative budget and the downward sensible heat flux. The latter two factors overwhelmed the heat loss due to sublimation and evaporation and thus provoked snow melting. It should be noted that in order to make more general conclusions on the effect of foehns on snow melt and glacier mass balance a much longer timeseries of observations has to be considered.

**3 Conclusions**

We presented the detailed analysis of the episode of an easterly flow over Svalbard which allowed us to investigate the main features of foehn and of the associated orographic flows. This was possible due to the availability of multiplatform observations obtained during the ACLOUD/PASCAL campaigns. This analysis revealed a strong resemblance of the considered foehn episode to other foehns, especially to foehn over the Antarctic Peninsula (Elvidge et al. 2015, Elvidge and Renfrew 2016,

Elvidge et al. 2016). The following results are obtained in this study:

1.    Downslope windstorm on the western slopes of Svalbard at the end of May 2017 appeared in the form of foehn, that is, there was a significant increase in air temperature and a decrease in relative humidity on the lee side of the mountains. A typical foehn clearance in could cover was observed. Wind speed amplification during the downslope windstorm was observed over the leeward slopes or near the foot of the slopes. Foehn wind near the surface did not propagate downstream. On the

contrary, the high-temperature and low-humidity effect of foehn reached far downstream into the sea, to a distance of more than 100 km from the coast (Fig.7a).

2.    Foehn led to a significant transformation of the boundary layer; the height and stratification of the boundary layer was largely determined by the downslope wind dynamics. A well-mixed layer was observed on the downwind side and over the fjords of western Svalbard. We suggest that it might have been produced both by the isentropic air descent and by the

increased turbulence due to wave breaking. Below this well-mixed layer, a shallow stably stratified boundary layer formed



over the snow surface and especially over the cold open sea. Foehn warming on the downwind side of Svalbard increased the horizontal temperature gradient across the sea ice edge in the Fram Strait. The temperature contrast between the northernmost profile over sea ice and in the lee of Svalbard reached 15 K. Baroclinicity associated with the horizontal temperature gradient might have affected the strength of the tip jet which formed at the northern tip of Svalbard.

3.        Gap winds, as well as tip jets, observed during the considered episode, differed from foehn by their larger horizontal extent (due to the absence of a hydraulic jump) and by almost complete absence of warming in the surface layer (since the vertical air displacements were small). However, above the surface layer, the air was significantly warmer and drier, since the foehn effect spread over the whole leeward region of Svalbard.

4.        Numerical modelling showed that the wind regimes of foehn and gap flows were determined by the flow direction
and Froude number ($Fr$) in the incoming flow. Downslope wind amplification as well as gap flows were most prominent when $Fr$ was less than 0.8, although for greater $Fr$ downslope windstorm could propagate slightly further from the lee slope.

5.        Foehn caused an intensification of the surface-atmosphere heat exchange in the Ny-Alesund and Longyearbyen (Adventdalen valley) regions, which accelerated the snow melt and resulted in an almost complete disappearance of snow cover at the Adventdalen station. A large vertical gradient of specific humidity in the surface layer resulted in large values of
latent heat flux. The latter formed due to the presence of snowmelt water and the air drying induced by foehn. The increased sensible heat flux directed from the atmosphere towards the surface partly compensated air cooling due to evaporation. Thus, in accordance with Elvidge et al. (2016), melting during the foehn occurred primarily due to the large positive short-wave radiation budget and, secondly, due to the downward sensible heat flux, caused by the foehn warming.

To conclude, the presented analysis revealed the detailed structure of the foehn effect, downslope windstorms, gap flows and
tip jets during easterly flow over Svalbard and their dependence on the incoming flow characteristics. The demonstrated acceleration of snowmelt during foehn suggests that the frequency of occurrence of an easterly flow in spring/summer might affect the glacier heat and mass budgets on the western side of Svalbard.

**Appendix A: Underestimation of air temperature in WRF simulations**

WRF consistently underestimated temperature (mean bias is -2.9 K) in the lower layers (surface and boundary layers)
compared with observational data (Fig.A1, 5), however, the model satisfactorily reproduced thermal stratification in most locations. This could be due to erroneous mixing in the boundary and surface layers in the model, however additional simulations with another boundary-layer parametrization (Asymmetric Convection Model 2, which is a hybrid non-local scheme in contrast to local MYNN scheme) also failed to simulate surface temperature. Moreover, the model underestimated temperature not only at those stations where it incorrectly reproduced the wind regime (for example, in Ny-Ålesund, the model
erroneously simulates wake, Fig.A1, 7), but also at stations where the wind was reproduced correctly (e.g. Pyramiden, Adventdalen, Barentsburg, Fig.A1). Cloud cover was rather well reproduced by the model, therefore, the differences in the incoming radiation between the observations and the model were small (not shown), and this could not be the reason for the



systematic temperature underestimation. We think, however, that these model difficulties do not affect our main conclusions because, e.g. the reproduction of stability is more important than the absolute values of temperature.


## Appendix B: Incoming flow

The incoming flow is assumed to be unperturbed by the orography (i.e., without the influence of orographic waves and upstream blocking) on the windward side of an obstacle. To analyze the incoming flow, we used the profiles of wind speed and temperature (Fig.B1) in 80˚N, 35˚E (near Kvitøya) (point 1 on Fig.2), far enough from Svalbard not to be affected by

blocking. To calculate the Froude number, we used $U$ and $N$ averaged in the lowest 1.7-km (maximum height of the mountain ridges) layer and the prevailing mountain ridge height $h$ was set as 1000 m. There is no consensus on over which layer to average U and N to calculate the Froude number; however, our previous study (Shestakova and Moiseenko, 2018) showed that the use of an average or maximum $U$ did not significantly affect the results.

**Data availability**

Surface meteorological observations in Ny-Alesund are available in Maturilli (2018b). Surface radiation observations in Ny-Alesund are available in Maturilli (2018a). Radiosounding observations in Ny-Alesund are available in Maturilli (2017). Radiosounding observations on RV Polarstern are available in Schmithüsen (2017). Polar 5 drop sondes observations are available in Ehrlich et al. (2019a). Aircraft observations during ACLOUD are available in Ehrlich et al. (2019b). Surface

meteorological and radiation observations in Adventdalen are available on the UNIS (the University Centre in Svalbard) website: https://www.unis.no/resources/weather-stations/ (last access: 22 April 2021). Archive meteorological data for other stations in Svalbard was obtained from the Yr website (a joint service by the Norwegian Meteorological Institute and the Norwegian Broadcasting Corporation): https://www.yr.no/ (last access: 22 April 2021). GFS FNL analysis was obtained from the Research Data Archive (RDA), managed by the Data Engineering and Curation Section (DECS) of the Computational and

Information Systems Laboratory (CISL) at the National Center for Atmospheric Research: https://rda.ucar.edu/datasets/ds083.2/ (last access: 22 April 2021). The YOPP analysis of ECMWF (European Centre for Medium-Range Weather Forecast) was obtained from the ECMWF website: https://apps.ecmwf.int/datasets/data/yopp/levtype=sfc/type=cf/ (last access: 22 April 2021).

**Author contributions**

The main part of the paper was prepared by AAS and DGC. Data were provided by JH and MM. AAS, DGC, CL and MM discussed the results and contributed to the writing of the paper.

**Competing interests**

The authors declare that they have no conflict of interest.



**Disclaimer**

The contents of this paper are solely the responsibility of the grantee and do not necessarily represent the official views of the supporting agencies.

**Acknowledgements**

The authors would like to acknowledge the ACLOUD/PASCAL campaigns participants for collecting the observational data used in this study. We acknowledge the use of imagery from the NASA Worldview application (https://worldview.earthdata.nasa.gov), part of the NASA Earth Observing System Data and Information System (EOSDIS).

**Financial support**

We gratefully acknowledge the funding by the Deutsche Forschungsgemeinschaft (DFG, German Research Foun-dation) project number 268020496 TRR 172 within the Transregional Collaborative Research Center ArctiC Amplification: Climate-Relevant Atmospheric and SurfaCe Processes, and Feedback Mechanisms (AC)3. The analysis of the orographic wind dynamics, WRF modelling and the calculations of the surface heat budget terms were funded by the Russian Science Foundation grant number 18-77-10072.

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






Table 1. Observations during foehn episode in 2017

| Type of Observation | Location | Temporal resolution | Temporal coverage | Vertical resolution / height of measurements |
|---|---|---|---|---|
| Surface meteorological observations | Ny-Alesund Barentsburg Adventdalen Longyear and other stations (Fig.1) | 1 min 3 h 1 s, 20 Hz 1 h | Whole episode | Main meteorological parameters: 2 m AGL, Wind: 10 m AGL |
| Surface radiation | Ny-Alesund | 1 min | Whole episode | 2 m AGL |
| Gradient masts | Ny-Alesund Adventdalen | 1 min 1 s | Whole episode | 2 m and 10 m (except relative humidity) AGL 2 m and 10 m AGL |
| Ship in-situ observations | See Fig.1 | 1 min | 12:00 May 30 – 12:00 May 31 | Main meteorological parameters: 29 m AMSL, Wind: 39 m AMSL |
| Radiosounding | Ny-Alesund | 6 h | Whole episode | Vertical resolution~ 5 m |
| Ship radiosounding | See Fig.1 | 6 h | 12:00 May 30 – 12:00 May 31 | Vertical resolution~ 20-30 m |
| Aircraft in-situ observations | Polar 5 Polar 6 (see Fig.1) | 1 s | ~ 15:00-19:00 May 31 | Height of observations: 0-3.7 km |
| Drop sondes (Polar 5) | See Fig.1 | - | May 31 (18:11, 18:21) | Vertical resolution ~ 5-10 m, height of observations 0-3.3 km |

Table 2. Melting and evaporation rates in Ny-Alesund and Adventdalen, calculated from the heat budget

| | Melting rate, mm/day w.e. | | Evaporation rate, mm/day | |
|---|---|---|---|---|
| | Ny-Alesund | Adventdalen | Ny-Alesund | Adventdalen |
| May 29 | 0.4 | 7.8 | 0.4 | 1.5 |
| May 30 | 6.2 | 16.4 | 0.8 | 3.1 |
| May 31 | 8.9 | 33.2 | 0.4 | 3.4 |




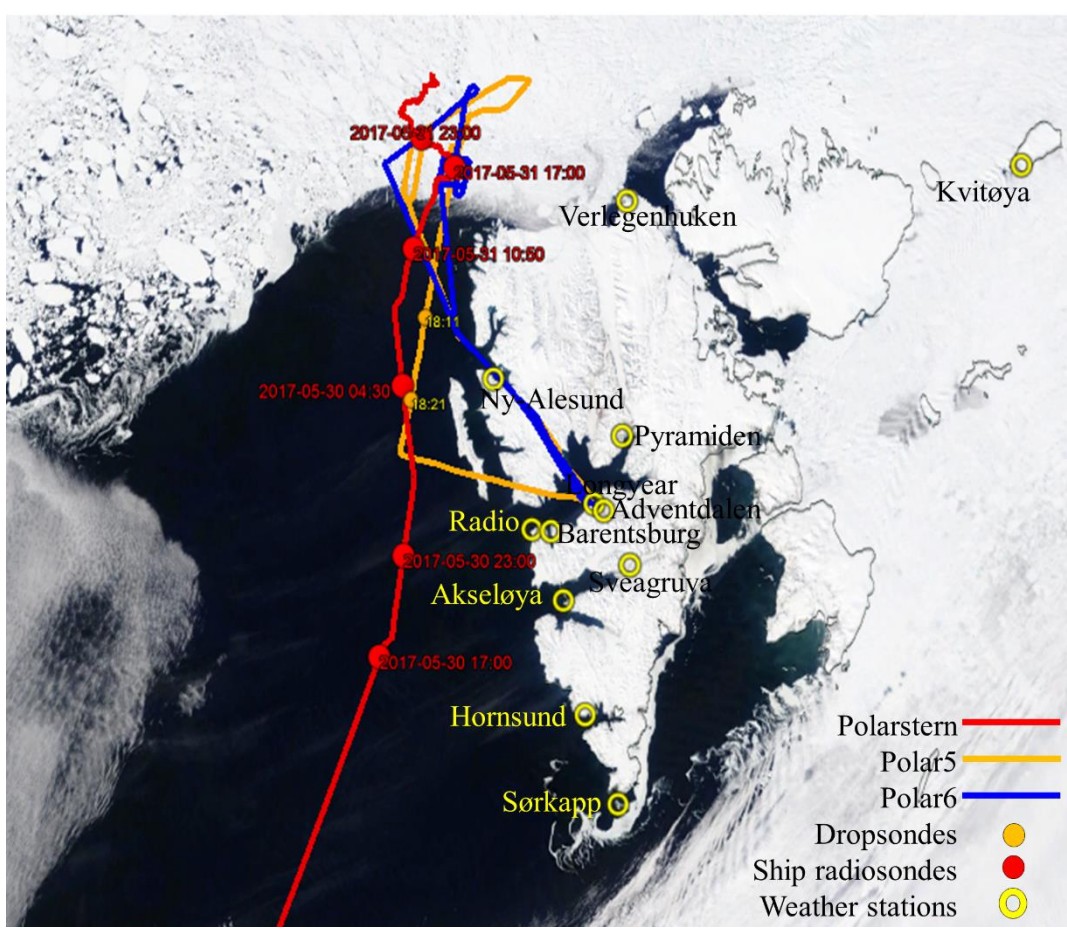

**Figure 1**: Satellite visible image (Modis corrected reflectance from NASA Worldview, https://worldview.earthdata.nasa.gov) of the study area at ~12 UTC May 31, 2017. Lines and points show ship and aircraft tracks, locations of dropsondes and radiosondes launches and weather stations in Spitsbergen




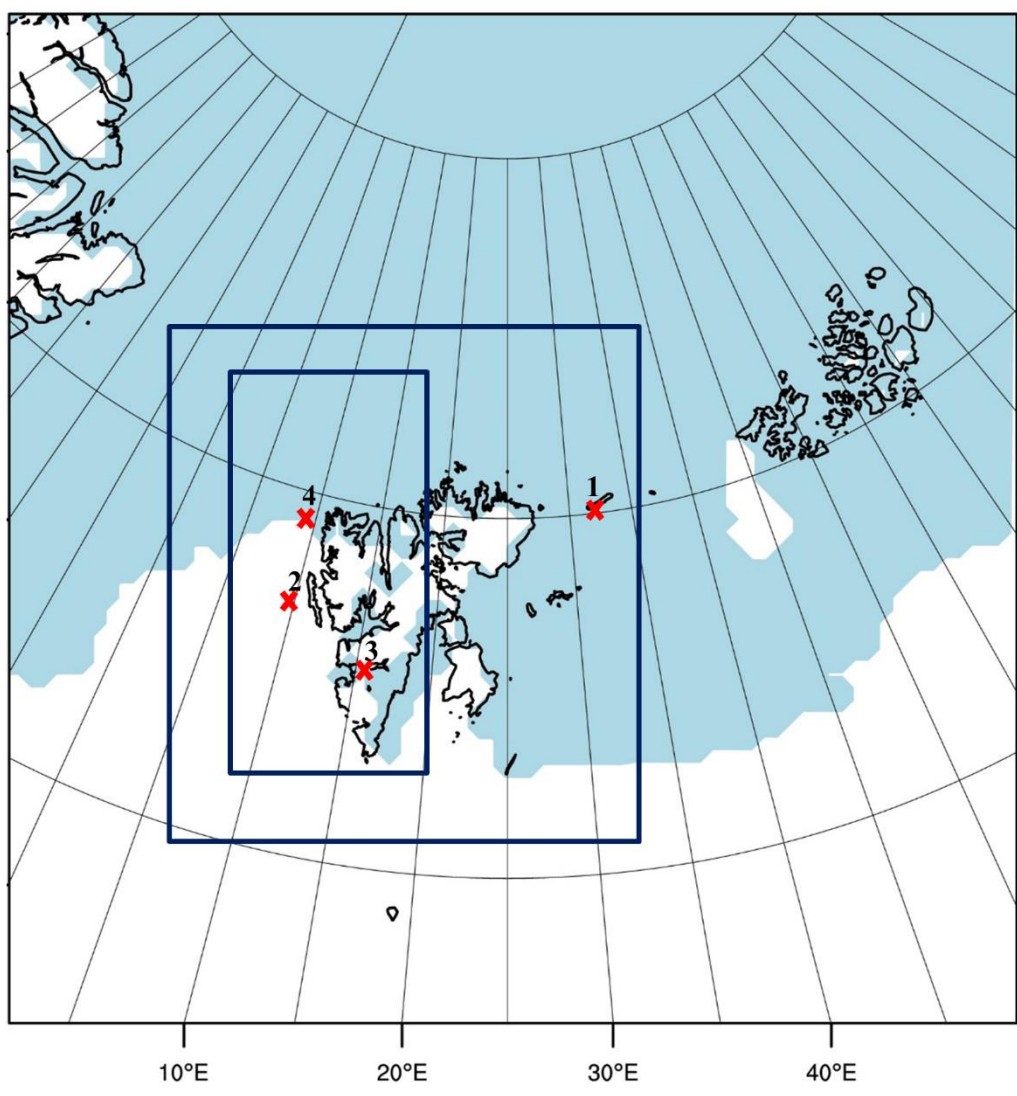

**Figure 2:** WRF outer domain 1 and nested domains 2 and 3. Sea ice is shaded. Red X-marks show points where vertical profiles were analyzed in Figure 9





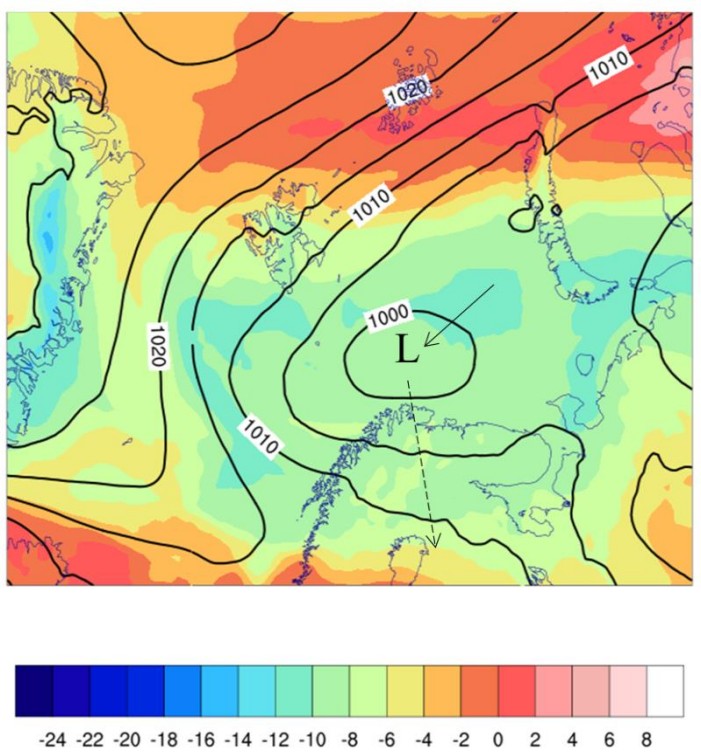

**Figure 3:** Synoptic chart at 06 UTC 30 May 2017 according to ECMWF YOPP analysis: sea-level pressure (black contours, every 5 hPa) and air temperature at 850 hPa level (color, every 2 K). Arrows show movements of cyclone center during May 29-31




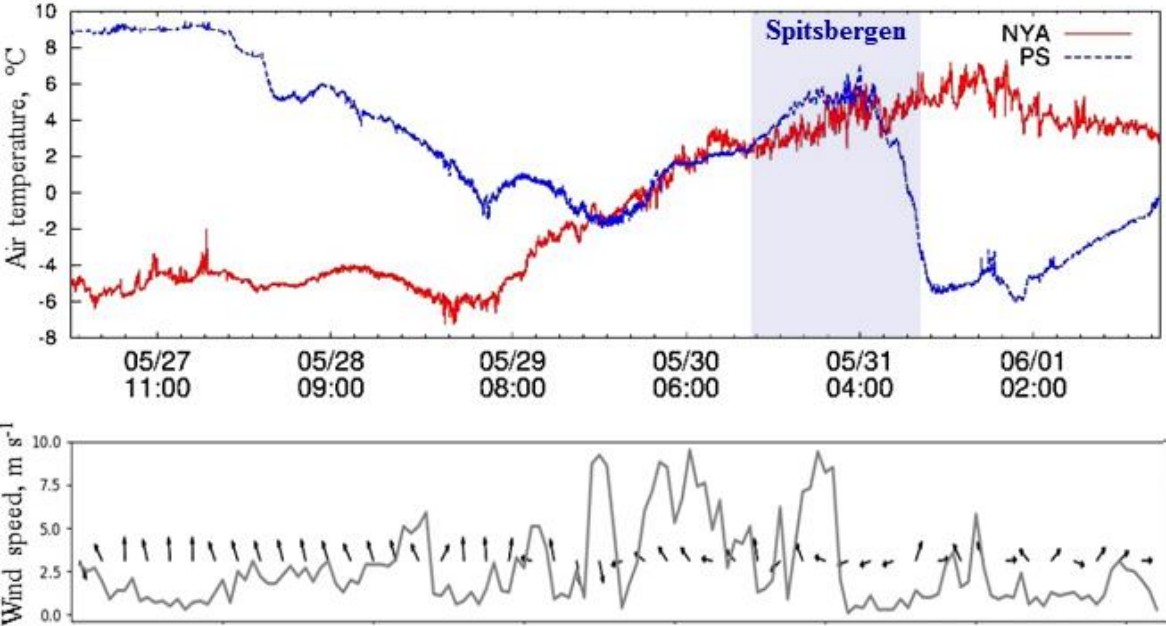

**Figure 4:** a) Time series of air temperature in Ny-Alesund (NYA) and on RV Polarstern (PS) as shown similarly by Knudsen et al. (2018). Blue shading shows the time period when Polarstern was downwind from Spitsbergen. b) Time series of wind speed (line) and direction (arrows) in Ny-Alesund



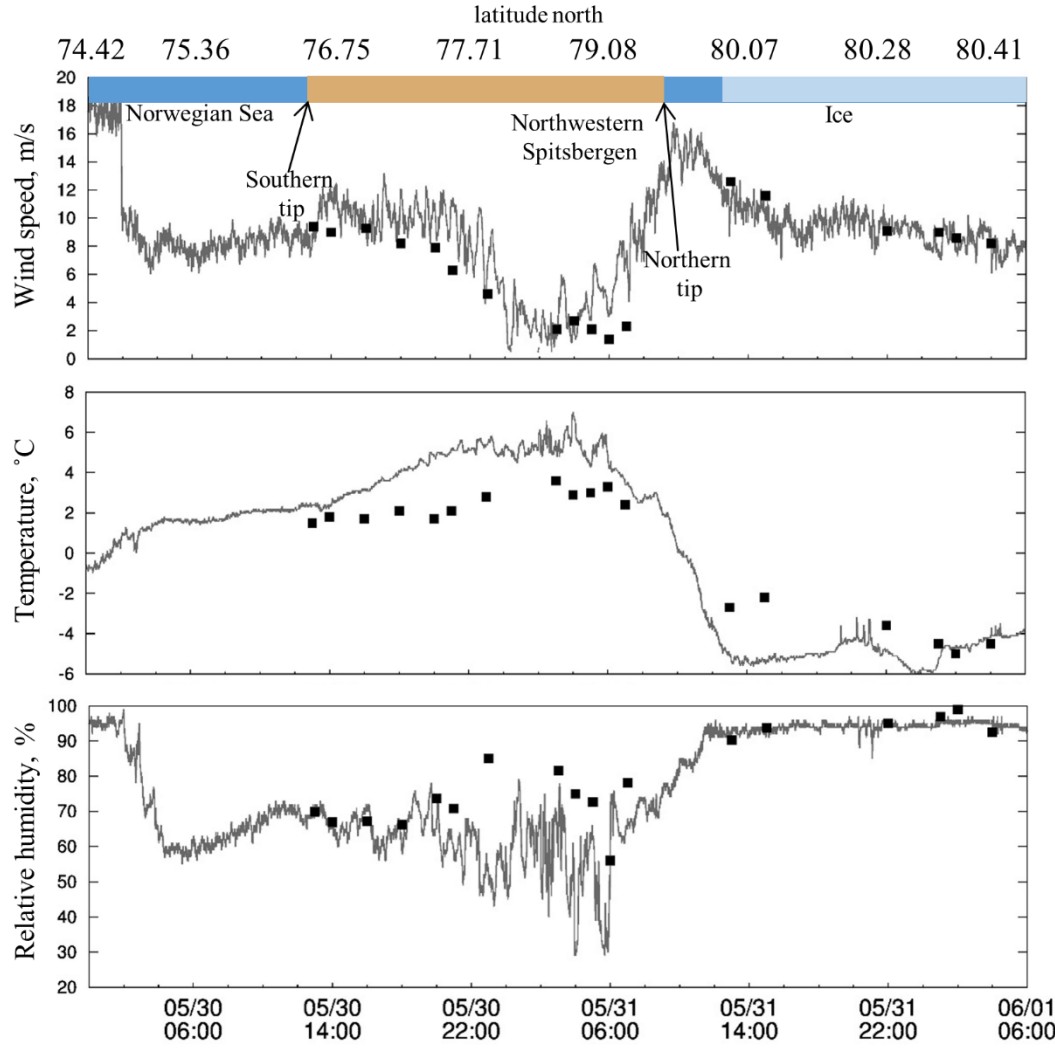

**Figure 5:** Wind speed (top), air temperature (center) and relative humidity (bottom) measured on RV Polarstern on 30-31 May 2017. Black squares show modelling data at the grid points nearest to the Polarstern track. Color panels on the top show when RV Polarstern was in the open sea (blue), downwind from Svalbard (brown) and in the sea ice (light blue)



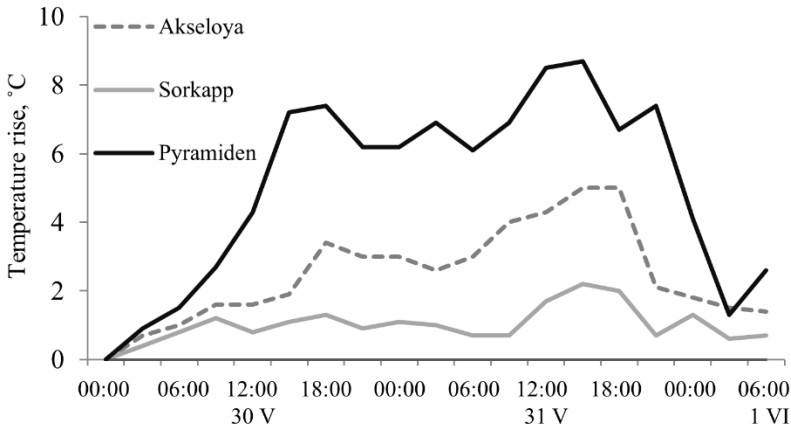

**Figure 6:** Temperature rise (observed temperature "minus" temperature at the initial time (00 UTC 30 May)) during foehn episode according to AWS observations


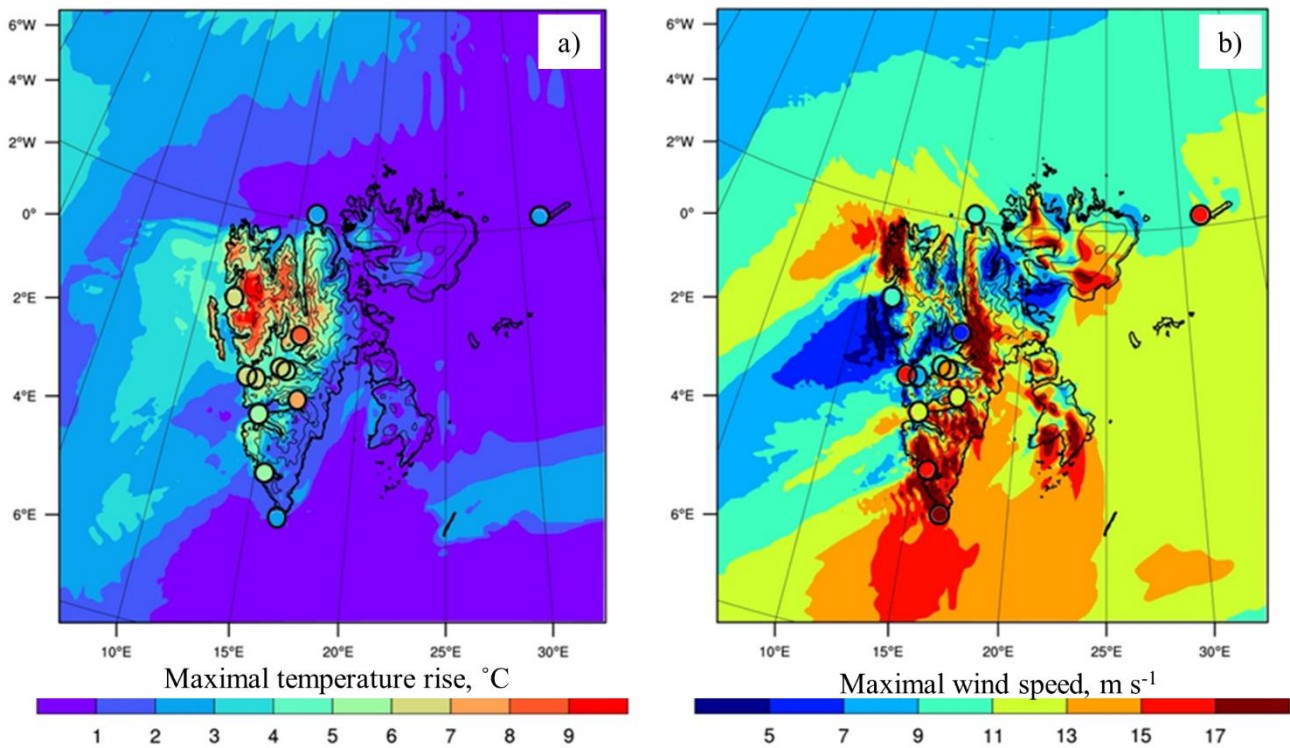

**Figure 7:** Maximal temperature rise at 2 m AGL (see explanations in the text) (left) and wind speed at 10 m AGL (right) during episode 2017 according to modelling and observations at weather stations (circles)










**Figure 8:** Wind speed (left column) and temperature (right column) maps (a, b) and cross-sections along the northern line Y-Y' (c, d) and the southern line Z-Z' (e, f) at 12 UTC May 30, 2017 according to modelling results. Arrows show wind vectors; red dots in the wind speed cross-sections mark regions with Ri<0.25


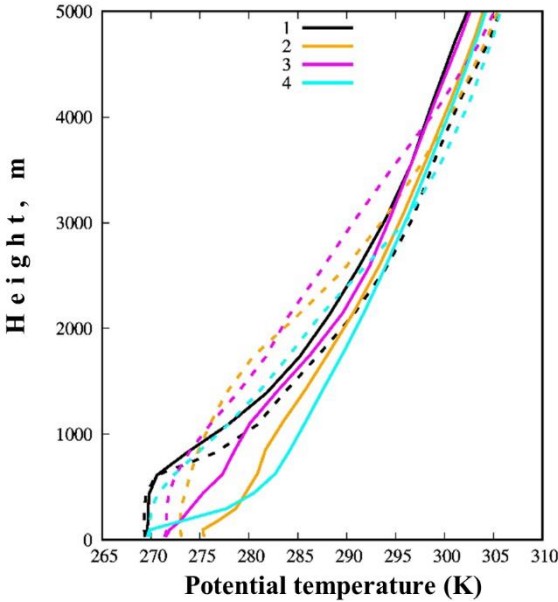

**Figure 9:** Vertical profiles of potential temperature in points 1-4 (see Fig.2) upstream and downstream of Spitsbergen at the beginning (06 UTC 30 May 2017, dotted lines) and in the middle of foehn episode (06 UTC 31 May 2017, solid lines) according

to modelling results

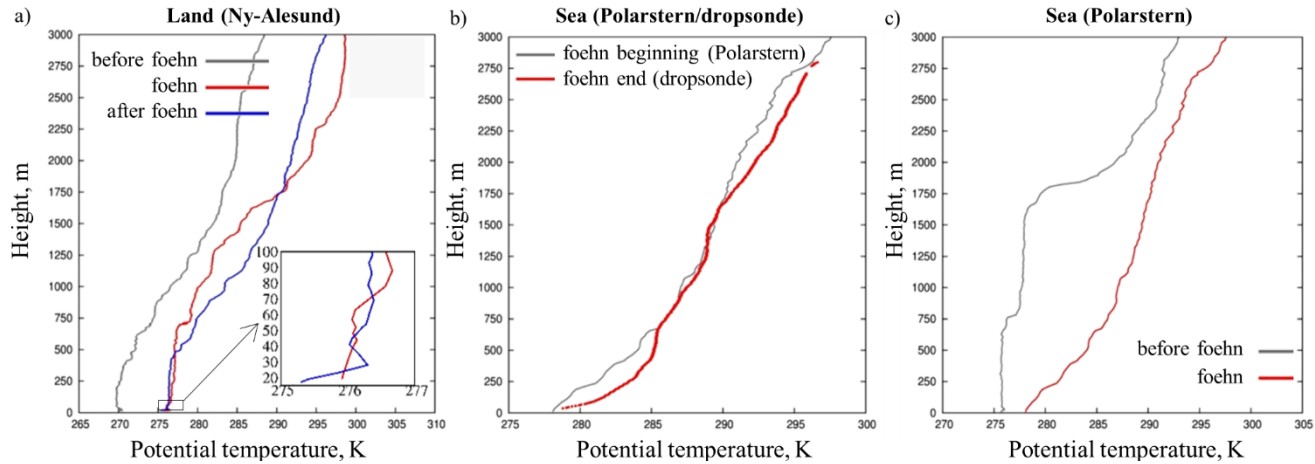





**Figure 10:** Observations of potential temperature in the lower troposphere a) over land (Ny-Alesund) and b), c) over sea. a)
Radiosonde observations over Ny-Alesund before foehn onset (12 UTC May 29), during foehn (12 UTC May 30) and after
foehn episode (12 UTC June 1). The lowermost 100-m layer is zoomed (see inset). b) Polarstern radiosonde observations at
04:34 May 31 and dropsonde observations at 18:21 May 31. c) Polarstern radiosonde observations south from Spitsbergen
(and thus before foehn, at 11:00 May 30) and in front of north-western Spitsbergen (during foehn, at 4:34 May 31)

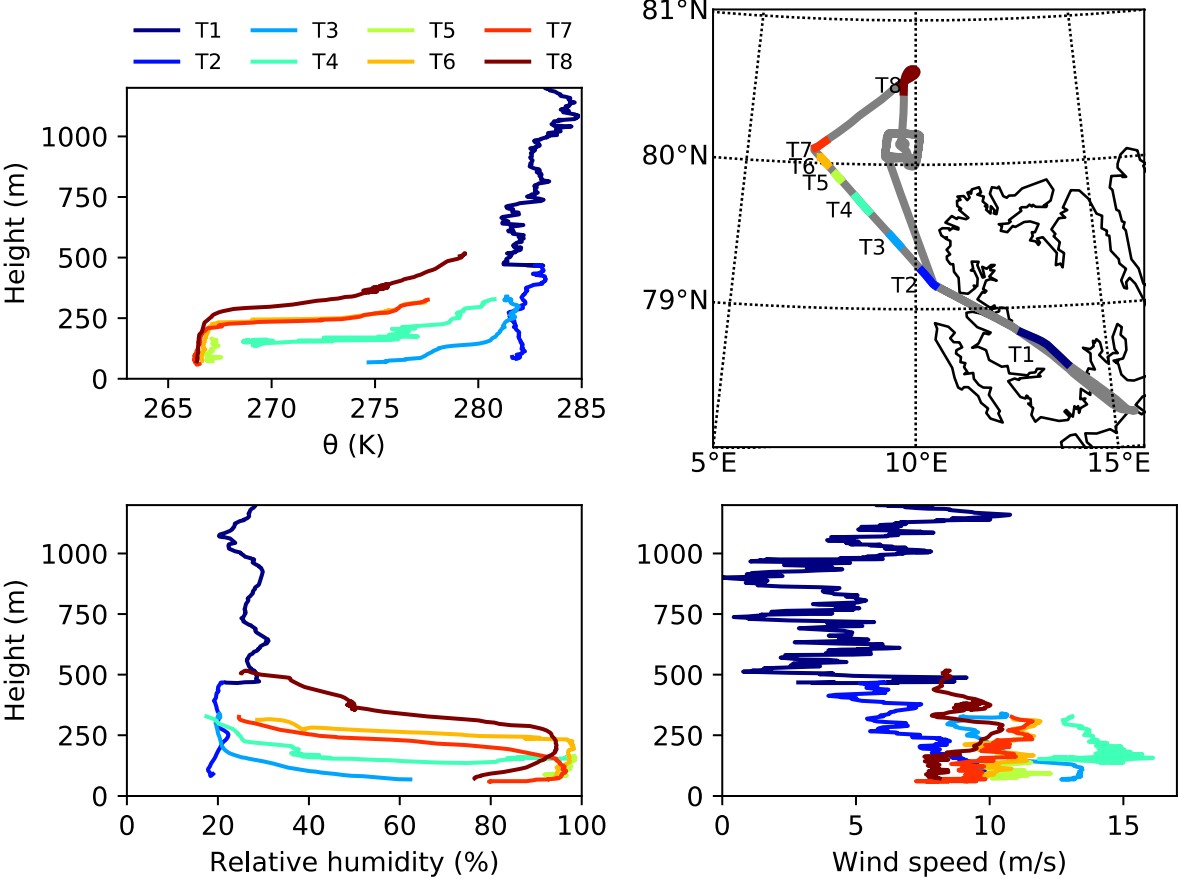

**Figure 11:** Vertical profiles of potential temperature (left top), relative humidity (left bottom) and wind speed (right bottom)
as observed by the Polar 6 aircraft along its track line (right top).

**Figure 12:** Dynamics of orographic flows during the observed episode 2017: wind speed (at 10 m AGL) normalized to windward wind speed (averaged in the lowest 1.7-km layer in 80˚N, 35˚E (near Kvitøya)) (color) and wind direction (barbs) at 10 m AGL and Fr number in the incoming flow (in the center) according to WRF modelling



**Figure 13:** Terms of radiation and heat budget in Ny-Alesund (left) and Adventdalen (right) during 29 May – 1 June 2017: a)-b) shortwave (orange line) and longwave (magenta line) radiation budget, albedo (grey line) and residual term of heat budget (black line), c)-d) turbulent heat fluxes (sensible (red boxes) and latent (blue boxes)) and relative humidity (green line) at 2 m (c) and 10 m AGL (d), e)-f) wind speed (black line), direction (points) at 10 m AGL and surface temperature (cyan line).



**Figure A1:** WRF model verification against AWS (a-d) and radiosonde data in Ny-Alesund (e) and Polarstern (f). a)-d) 10-m wind speed *V*, 2-m temperature *T* and wind direction *Dir* according to AWS observations and WRF simulation. Wind speed at Adventdalen station was measured with sonic anemometer at 2.7 m AGL. e)-f) Wind speed (left) and temperature (right) according to observations (solid line) and WRF simulation (squares).

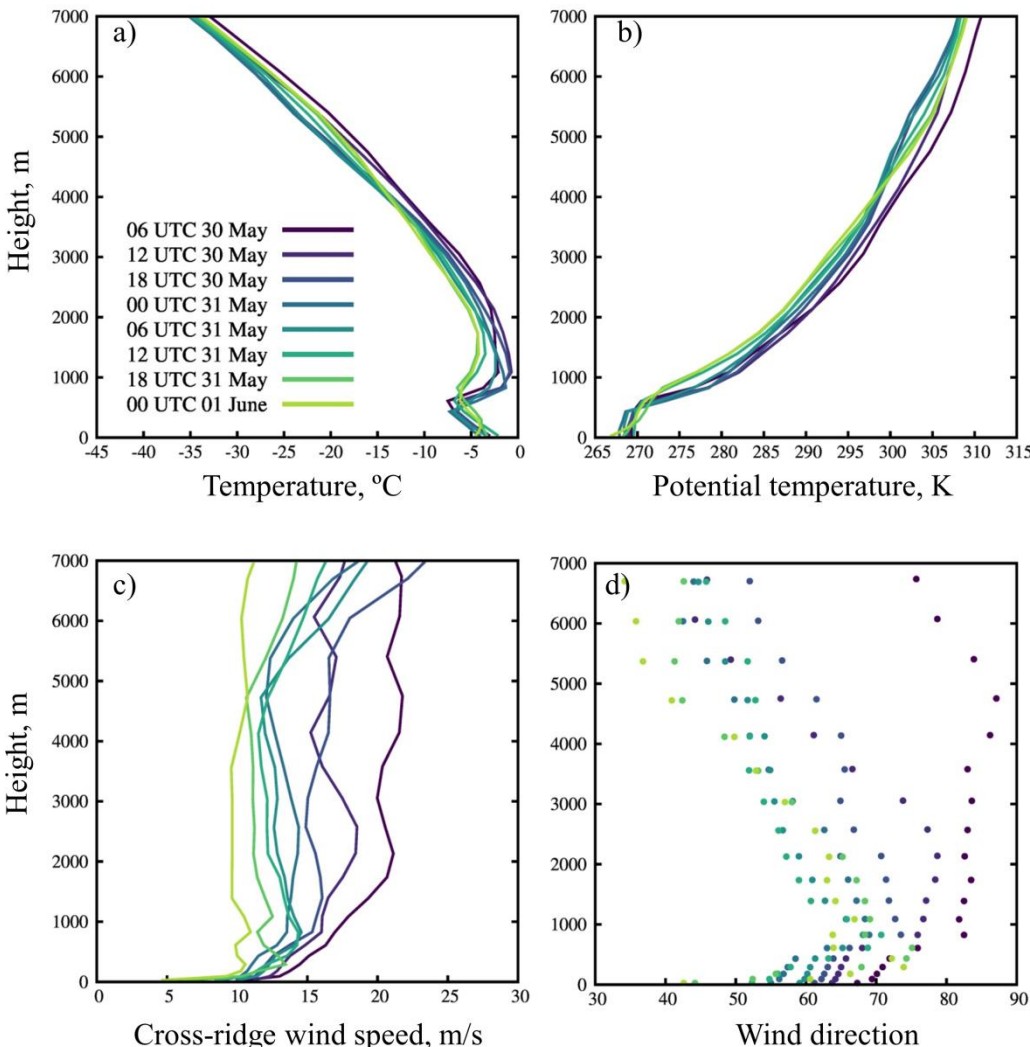

**Figure B1:** Vertical profiles of incoming flow during episode 2017 according to WRF modelling: a) air temperature, b) potential temperature, c) cross-ridge wind component and d) wind direction

775