# Peer review of "The foehn effect during easterly flow over Svalbard"

_Atmospheric Chemistry and Physics, 2021_

## Referee Comment (RC1)

Review of **"Foehn effect during easterly flow over Svalbard"**
by Shestakova, Chechin, Lüpkes, Hartmann and Maturilli, ACP

The authors document a case study of a well-defined foehn event from May 2017 over Svalbard. The event is described using an impressive array of observational platforms, including flight-level and dropsonde observations from an aircraft, surface-layer and radiosonde observations from a research vessel and surface-layer and radiosonde observations from Svalbard. The observations are complemented by output from WRF simulations.

Overall, the paper presents the most comprehensive description and analysis of a foehn event over Svalbard that I have seen, and nicely expands the locations where foehn events have been well documented. The case is compared to those found over the Antarctic Peninsula and this comparison is useful and appropriate. Overall, this is a carefully conducted study with several important findings. It is generally well written, and the quality of the presentation is excellent. I have some minor comments and suggestions, which are generally aimed at improving the presentation.

**Specific Figure suggestions:**

**Figure 1 -** Are all the place names used in this figure?

**Figure 4** – I wasn't sure this figure was necessary for the paper. The Polarstern is moving a lot during this period and a shorter time series is given in Fig 5. The winds at Ny Alesund show the foehn period, but also show elevated wind speeds on 29 May, which aren't really discussed. Not sure this is needed.

**Figure 6** – there is a bit of a mismatch between the names on lines 190-191 and the names in Figure 6. "Verkegenhuken" is not shown on Fig 6. You could consider showing the upstream temperature from Kvitøya on Figure 6, and changing upsteam to be blue, downstream to be red. Maybe rephrase caption, "temperature change from 'initial' time at each individual AWS station, where the initial time is 00 UTC 30 May".

**Figure 7** – I'm not sure what wind speed is plotted here? Is it a 'maximum' (better than 'maximal') wind speed?

**Figure 9** – I think this could be more clearly presented. Add a location map as another panel. Maybe choose colours so 'cold' colour is upstream and 3 'warm' colours are the 3 downstream locations.

**Figure 10** – The layout here is a little confusing. a) and b) compare two soundings in approximately the same location. But c) compares two soundings in different locations and one of them is the same sounding as panel b). I'd be tempted to merge (b) and (c) – but make it clear in the caption that the first sounding is in a different location. I'd also redo this figure with a different colour scheme, so first profile blue, then foehn and after-foehn profiles as red and magenta (as 'warm' colours). Rephrase caption to be clear.

**Figure 11 –** I'd recommend adding sea-ice concentraton to 11b.

**Figure 13** – I'd recommend adding a zero line to the bottom panels. Caption should mention these are net SW and net LW. I'd move 'albedo' to end of the sentence about top panels, as it is the right hand axis.

**Figure A1** – The caption needs improving. Make it clear that observations are solid line and WRF simulations are dotted lines and squares for WD? You comment on fact tht WRF underestimates the air temperature. But is is also poor for wind speed at Ny Alesund for some of this period. You maybe want to comment on that? I guess it is an area of complex orography so 10-m wind speed is challenging.

**Minor Suggestions**

Line 1 – I'd suggest adding a "**The** foehn effect.." to the title.
L17 – "downwind of Svalbard"
L19 - "A positive… budget at the surface…"
L30 – "Altogether, this results in the highest…Europe being observed in …"
L40 – delete "the"
L53 – could also cite Elvidge et al. (2020) here, this is a relatively new paper which focuses on surface energy budget over the Antarctic Peninsula; and also Turton et al. (2018) which used AWS observations to investigate foehn winds in this area.
L69 - I'd rephrase as southern Greenland tip jets (plural) because there are both westerly jets (Doyle and Shapiro 1999) and easterly tip jets (e.g. Renfrew et al. 2009; Outten et al. 2009).
L76 – it may be pertinent to cite a more general paper, such as a review paper, when discussing hydraulic jumps, e.g. Durran (1990); Smith (1989).
L77 – the horizontal pressure gradient is down the slope. not along it.
L103 – delete "used" and, edit to be "section 2.1 and the setup… "The synoptic background…" Then each sentence is about each section.
L117 "Information about the observations…"
L121 "a series of …"
L166 "reached a maximum"
L199 – I'd rephrase "Obviously" as we are still at the beginning of the paper and you haven't presented evidence that it is obvious to the reader yet.
L217 – "over southern Svalbard"
L232 – start a new paragraph here, with "The vertical…
L261 – you cite "profile 1 in Fig 9" here, but that is the upwind profile, did you mean to cite another (downwind) profile? Incidentally, I don't think you really need Figure 2 – it is not that useful. Instead I'd consider just plotting the domains 2 and 3, as a second panel for Fig 9, so that it is easier to see where these profiles are located when looking at Fig 9.
L294 – "the North"
L295 – I would rephrase, the diagram doesn't "clearly show the downward propagation" because it is a snapshot in time.
L298 – you note the sea-ice here – I think it would be helpful to mark sea-ice concentration on Fig 11b.
L301 – "downwind of the mountains"

L303 – rather than "advective origin" perhaps you mean "large-scale flow" or something?

L306 – I'd rephrase as "increase of the BL height to the North" – so it is not ambiguous

L319 – delete "a"

L286 – I'd be clear that you are plotting **net** SW radiation

Figure 13 – I'd mark the zero line on the bottom panels, it would enable easy comparison of T_surface against 0.

L422 – "was large"

L443 – "conclude" is the wrong word here, because there is an element of hypothesis here, so I would say "we surmise"

L458 – "cloud"

**References**

Durran, D. R. (1990). Mountain waves and downslope winds. In *Atmospheric processes over complex terrain* (pp. 59-81). American Meteorological Society, Boston, MA.

Elvidge, A. D., P. Kuipers Munneke, J. C. King, I. A. Renfrew, E. Gilbert 2020: Atmospheric drivers of melt on Larsen C Ice Shelf: surface energy budget regimes and the impact of foehn, *J. Geophysical Research: Atmospheres,* **125,** e2020JD032463. doi:10.1029/2020JD032463

Outten S.D., I.A. Renfrew, and G.N. Petersen, 2009: An easterly tip jet off Cape Farewell, Greenland. Part II: Simulations and dynamics, *Quarterly J. Royal Meteorol. Soc.,* **135,** 1934-1949.

Renfrew, I.A., S.D. Outten and G.W.K. Moore, 2009: An easterly tip jet off Cape Farewell, Greenland. Part I: Aircraft observations, *Quarterly J. Royal Meteorol. Soc.,* **135,** 1919-1933.

Smith, R. B. (1989), Hydrostatic airflow over mountains, *Adv. Geophys.,*31, 1–41, doi:10.1016/S0065-2687(08)60052-7.

Turton, J.V., Kirchgaessner, A.,Ross, A. N.,&King, J.C. (2018). The spatial distribution and temporal variability of föhn winds over the LarsenC Ice Shelf, Antarctica. *Quarterly Journal of the Royal Meteorological Society*, 144(713), 1169–1178. https://doi.org/10.1002/qj.3284

---

## Author Response (AR1)

Reviewer 1

We thank the reviewer for his comments that helped to improve the manuscript and the presentation. We tried to take into account all of the reviewer's suggestions, as explained in detail below.

«Figure 1 - Are all the place names used in this figure?»

Some of the AWS names are not used, but observations from all of these stations are used (for example, in Fig.5).

«Figure 4 – I wasn't sure this figure was necessary for the paper. The Polarstern is moving a lot during this period and a shorter time series is given in Fig 5. The winds at Ny Alesund show the foehn period, but also show elevated wind speeds on 29 May, which aren't really discussed. Not sure this is needed.»

Figure 4 was removed.

«Figure 6 – there is a bit of a mismatch between the names on lines 190-191 and the names in Figure 6. "Verkegenhuken" is not shown on Fig 6. You could consider showing the upstream temperature from Kvitøya on Figure 6, and changing upsteam to be blue, downstream to be red. Maybe rephrase caption, "temperature change from 'initial' time at each individual AWS station, where the initial time is 00 UTC 30 May".»

Fixed. The text describing the figure was slightly rewritten (lines 181-188).

«Figure 7 – I'm not sure what wind speed is plotted here? Is it a 'maximum' (better than 'maximal') wind speed?»

"Maximal" was replaced by "maximum".

«Figure 9 – I think this could be more clearly presented. Add a location map as another panel. Maybe choose colours so 'cold' colour is upstream and 3 'warm' colours are the 3 downstream locations.»

Fixed.

«Figure 10 – The layout here is a little confusing. a) and b) compare two soundings in approximately the same location. But c) compares two soundings in different locations and one of them is the same sounding as panel b). I'd be tempted to merge (b) and (c) – but make it clear in the caption that the first sounding is in a different location. I'd also redo this figure with a different colour scheme, so first profile blue, then foehn and after-foehn profiles as red and magenta (as 'warm' colours). Rephrase caption to be clear.»

Fixed.

«Figure 11 – I'd recommend adding sea-ice concentraton to 11b.»

Fixed.

«Figure 13 – I'd recommend adding a zero line to the bottom panels. Caption should mention these are net SW and net LW. I'd move 'albedo' to end of the sentence about top panels, as it is the right hand axis.»

Fixed.

«Figure A1 – The caption needs improving. Make it clear that observations are solid line and WRF simulations are dotted lines and squares for WD? You comment on fact tht WRF underestimates the air temperature. But is is also poor for wind speed at Ny Alesund for some of this period. You maybe want to comment on that? I guess it is an area of complex orography so 10-m wind speed is challenging.»

Fixed. A sentence about the poor wind in the model was added (lines 507-509).

Minor Suggestions

«Line 1 – I'd suggest adding a "The foehn effect.." to the title.»

Added.

«L17 – "downwind of Svalbard"»

Fixed.

«L19 - "A positive… budget at the surface…"»

Fixed.

«L30 – "Altogether, this results in the highest…Europe being observed in …"»

This sentence was removed.

«L40 – delete "the"»

Fixed.

«L53 – could also cite Elvidge et al. (2020) here, this is a relatively new paper which focuses on surface energy budget over the Antarctic Peninsula; and also Turton et al. (2018) which used AWS observations to investigate foehn winds in this area.»

These references were added.

«L69 - I'd rephrase as southern Greenland tip jets (plural) because there are both westerly jets (Doyle and Shapiro 1999) and easterly tip jets (e.g. Renfrew et al. 2009; Outten et al. 2009).»

Fixed.

«L76 – it may be pertinent to cite a more general paper, such as a review paper, when discussing hydraulic jumps, e.g. Durran (1990); Smith (1989).»

Fixed.

«L77 – the horizontal pressure gradient is down the slope. not along it.»

This sentence was removed.

«L103 – delete "used" and, edit to be "section 2.1 and the setup… "The synoptic background…" Then each sentence is about each section.»

Fixed.

«L117 "Information about the observations…"»

Fixed.

«L121 "a series of …"»

Fixed.

«L166 "reached a maximum"»

Fixed.

«L199 – I'd rephrase "Obviously" as we are still at the beginning of the paper and you haven't presented evidence that it is obvious to the reader yet.»

Fixed.

«L217 – "over southern Svalbard"»

Fixed.

«L232 – start a new paragraph here, with "The vertical…"»

Fixed.

«L261 – you cite "profile 1 in Fig 9" here, but that is the upwind profile, did you mean to cite another (downwind) profile? Incidentally, I don't think you really need Figure 2 – it is not that useful. Instead I'd consider just plotting the domains 2 and 3, as a second panel for Fig 9, so that it is easier to see where these profiles are located when looking at Fig 9.»

No, we referenced exactly the incoming flow (upwind profile). Figure 2 was remade and placed as a second panel in Fig.9 (now Fig.7)

«L294 – "the North"»

Fixed.

«L295 – I would rephrase, the diagram doesn't "clearly show the downward propagation" because it is a snapshot in time.»

Fixed.

«L298 – you note the sea-ice here – I think it would be helpful to mark sea-ice concentration on Fig 11b.»

Sea-ice concentration was added.

«L301 – "downwind of the mountains"»

Fixed.

«L303 – rather than "advective origin" perhaps you mean "large-scale flow" or something?»

Rephrased.

«L306 – I'd rephrase as "increase of the BL height to the North" – so it is not ambiguous»

Rephrased.

«L319 – delete "a"»

Fixed.

«L286 – I'd be clear that you are plotting net SW radiation»

Fixed.

«Figure 13 – I'd mark the zero line on the bottom panels, it would enable easy comparison of T_surface against 0.»

Added.

«L422 – "was large"»

Fixed.

«L443 – "conclude" is the wrong word here, because there is an element of hypothesis here, so I would say "we surmise"»

Fixed.

«L458 – "cloud»

Fixed.

Reviewer 2

We thank the reviewer for his useful comments. Please, find the answer to each of the comments below.

«Although the paper is generally well written, easy to understand and follow, the presentation of the material seems rather lengthy and redundant in some parts. There is a long introduction that focuses on climate impacts and Arctic amplification. Well, that's the framework, but the paper doesn't make a conclusive contribution to the issue.»

The introduction was shortened.

«And then there is the leeward warming, which is looked at in too much detail for my understanding, since it is a known phenomenon. A novel aspect would be for the authors to present a Lagrangian analysis of the temperature and moisture evolution of air parcels starting upstream. This approach could really help to distinguish between the large-scale warming and the mesoscale effect due to the flow past the mountains. Indeed, the mesoscale numerical simulations are available and could easily be used to conduct such an analysis.»

and

«line 170-172: this is one of the too speculative sentences that could be strengthened by a more detailed analysis (the Lagrangian analysis referred to above) and its subsequent incorporation of the results into the content of the paper»

Elvidge et al. (2016) applied Lagrangian analysis to several cases of foehns over the Antarctic Peninsula and it helped them to distinguish the mechanisms of warming. This is the advantage of this approach and, indeed, Lagrangian analysis can be very useful in analyzing various episodes of Svalbard foehns. But for the episode under consideration, isentropic warming seems to be the only mechanism, since the air mass was dry and there was little precipitation (lines 264-266). Turbulent fluxes of heat and moisture could mix warmer and drier air from the upper layers into the Lagrangian volume. At the same time, turbulent exchange with the underlying surface would result in cooling and moistening of a Lagrangian volume. Quantitative estimates of the contribution of these fluxes to the evolution of the temperature and humidity of a Lagrangian volume would undoubtedly be useful, but we assume that there is a rather large uncertainty regarding the turbulent fluxes reproduced by the model in conditions of very complex orography. We consider the Lagrangian investigation for these reasons as an interesting goal, but it requires a separate future study. In the revised manuscript we explain this difference to the Lagrangian study of Elvidge et a. (2016) and hint to this as a future goal (lines 497-500).

«While browsing the literature on Svalbard, the reviewer came across a paper by Dörnbrack et al. (2010), who also studied a very similar episode of easterly airflow in this journal (Dörnbrack et al., 2010). Although they focused on aerosol distribution from airborne lidar measurements (incidentally using the POLAR 2 aircraft, a predecessor of the current AWI fleet used in the current study), they also discuss most of the phenomena studied in the current paper by Shestakova et al. (especially, the "orographic modification of the flow and of the atmospheric boundary layer during easterly flow over Svalbard." (line 94/95) as e.g. the formation of a "warm" wake in the lee of the mountains, the tipjet that forms on the northern edge of Svalbard, ....) . And there are also similarities in some of the diagrams: cf. current Fig. 7 in this paper and Fig. 9 in Dörnbrack et al. I think the authors should really try to better highlight the novelty of their analysis in a revised version of the paper.»

We thank the reviewer for mentioning the important reference to Dörnbrack et al. (2010) which we overlooked. In the revised version of the paper the study of Dörnbrack et al. (2010) is cited where appropriate, especially, concerning similar conclusions on the flow modification by the orography of

Svalbard. Of course, the paper (Dörnbrack et al. 2010) describes all the main features of the wind and temperature fields during foehn, which influence also the aerosol concentration. Also, such features are described in other articles (for example, Sandvik and Furevik, 2002; Skeie and Grønås, 2000; Reeve and Kolstad, 2011). However, in these articles, all these features are considered mainly based on modeling data or not in so much detail as in our study. In contrast, our focus is on studying the foehn effect based on in-situ observations (modeling is used only as an auxiliary tool for interpreting observations) and its impact on the boundary layer and heat balance in the surface layer. To our knowledge, such a large amount of data obtained from various observational platforms during the foehn event is presented for the first time and provides a more solid basis for quantification of the magnitude of the foehn effect, as well as of the orographic modifications of the air flow, such as gap flows and tip jets. Our research based on a completely independent data set confirms many of the features found in previous works (lines 66-68, 288-289, 492-496), and thus highlights these previous findings and shows the high degree of generality of these works.

Minor Comments

«lines 194: the Kvitoya station is missing in Figure 6; generally, the temperature rise increases already before, much earlier than in the time window as shown in Figure 6, see Figure 4; so, the respective discussion should be modified»

Figure 6 and the text (lines 181-188) have been revised. As could be seen from Fig.4 (now removed), the temperature rise began before May 30, and it was not associated with foehn (since the wind was first southerly and then northerly, but not easterly), but with large-scale advection. Since we are interested in foehn warming, we have chosen the period starting from May 30, when foehn began.

 «Figure 7: the labels on the left axes are misleading, since there is obviously no reference to the lines in the image»

 Fixed.

«Figure 9: dotted -> dashed»

 Fixed.

«line 416: Is the snow really melting there at these temperatures or it is mostly drifting away or evaporating?»

Indeed, the snow could sublime or melted water could evaporate, and this process is accounted for in the latent heat flux. But the estimated evaporation rate was one order smaller than the melting rate. The limitations of the estimation of melting rate in Adventdalen on May 31 are listed in lines 431-433.